# Outliers Matter—Correlation between S1 IgG SARS-CoV-2 Antibodies and Neutralizing SARS-CoV-2 Antibodies

**DOI:** 10.3390/microorganisms10102067

**Published:** 2022-10-19

**Authors:** Berthold Hocher, Anne Schönbrunn, Xin Chen, Bernhard K. Krämer, Volker von Baehr

**Affiliations:** 1Fifth Department of Medicine (Nephrology/Endocrinology/Rheumatology/Pneumonology), University Medical Centre Mannheim, University of Heidelberg, D-69120 Heidelberg, Germany; 2Institute of Medical Diagnostics, IMD Berlin-Potsdam, D-12247 Berlin, Germany

**Keywords:** SARS-CoV-2 virus, IgG antibodies against the S1 protein of the SARS-CoV-2 virus, neutralizing SARS-CoV-2 virus antibodies, correlation, clinical study

## Abstract

Vaccination against the SARS-CoV-2 virus or infection with SARS-CoV-2 will lead to the development of IgG antibodies against the S1 protein of the SARS-CoV-2 virus. However, even despite having high levels of IgG antibodies against the S1 protein of the SARS-CoV-2 virus, (re-)infection may occur. We thus examined 2994 consecutive blood samples of outpatients from the Berlin-Brandenburg area in Germany in which IgG antibodies against the S1 protein of the SARS-CoV-2 virus as well as neutralizing SARS-CoV-2 virus antibodies were determined from the same sample. When analyzing the entire study population (2994 outpatients), we saw that S1 IgG antibodies (women: 223.98 ± 3.81; men: 207.80 ± 4.59; *p* = 0.014) and neutralizing antibodies (women: 66.65 ± 0.82; men: 62.88 ± 1.01; *p* = 0.021) are slightly higher in women than in men. Curve fitting revealed a good non-linear relationship between S1 IgG and neutralizing SARS-CoV-2 antibodies. However, 51 out of the 2994 blood samples from individual subjects were positive with regard to the neutralizing antibodies and at the same time negative for S1 IgG antibodies, and 112 out of the 2994 blood samples from individual subjects were negative with regard to the neutralizing antibodies and at the same time positive for S1 IgG antibodies. In conclusion, our study shows that there is a relevant number of patients who, despite developing significant titers of S1 antibodies, do not have relevant amounts of neutralizing antibody titers and are probably at high risk of (re-)infection.

## 1. Introduction

Recent studies indicate a good correlation between the SARS-CoV-2 neutralizing antibody titer and clinical outcomes. For instance, it was shown that among fully vaccinated healthcare workers, the occurrence of breakthrough infections with SARS-CoV-2 was correlated with neutralizing antibody titers [1]. Different levels of neutralizing antibodies after full vaccination with different currently investigated SARS-CoV-2 vaccines correlate likewise very well with immune protection seen in clinical phase 3 studies [2]. Another study showed that the presence of neutralizing antibodies within the first weeks from the onset of symptoms correlates with time to a negative swab result, while the absence of neutralizing antibodies correlates with an increased risk of a fatal outcome [3]. Furthermore, the severity of COVID-19 disease correlates with the amount of circulating neutralizing antibodies [4]. Although antibodies against the S1 domain of the spike protein of the SARS-CoV-2 virus generally correlate well with neutralizing SARS-CoV-2 antibodies, the clinical value of S1 antibodies is less well established.

## 2. Methods

We examined consecutive blood samples of outpatients from the Berlin-Brandenburg area, Germany, in which IgG antibodies against the S1 protein of the SARS-CoV-2 virus [5] as well as neutralizing SARS-CoV-2 virus antibodies [6] were determined from the same samples. For a detailed description of the applied methods for detecting IgG antibodies against the S1 protein of the SARS-CoV-2 virus, see our recent methodological report [7].

Briefly, for quantitative detection of IgG against SARS-CoV-2 spike glycoprotein 1 (S1 subunit) an enzyme-linked immunosorbent assay (ELISA; EUROIMMUN) was used on a fully automated analyzer system (QuantiVac, EUROIMMUN, Lübeck, Germany) according to manufacturer’s instructions. The assay relies on six calibrators in order to quantify the IgG (S1)-concentration given as BAU/mL (Binding Antibody Units) and highly correlates with the “First WHO International Standard” (NIBSC code: 20/136). Values between 25.6 and 35.2 BAU/mL were considered borderline, while values above 35.2 BAU/mL were interpreted as positive.

SARS-CoV-2 sVNT Kit (cPAss from Genscript, Piscataway, NJ, United States) was used to evaluate the neutralizing capacity of anti-SARS-CoV-2 antibodies present in the serum. The method to evaluate neutralizing antibodies in this paper (ACE2-competition binding assay) mainly detects RBD-specific antibodies. This is a blocking enzyme-linked immunosorbent assay (ELISA), which mimics the virus-host interaction. The binding of a horseradish peroxidase-conjugated RBD-fragment of the SARS-CoV-2 (HRP-RBD) to the human host ACE2 receptor can be blocked by neutralizing antibodies against the SARS-CoV-2 spike protein, containing the RBD in the serum or plasma. The strength of the HRP signal indicates the degree of blockage and therefore indirectly the neutralizing capacity. The test (according to the manufacturer’s information and internal validations) recognizes neutralization against the wild-type variant as well as against the alpha, beta, and delta variants of the virus, but not against the omicron variant of the virus. The sVNT assay from Genscript has been validated and described previously [6,8,9,10,11].

## 3. Results

Blood from consecutive samples from 2994 outpatients (age: 55.2 +/− 16.9 years; 41.9% female/58.1% male) was taken between 1 April 2021 and 30 August 2021. Curve fitting revealed a good non-linear relationship between S1 IgG and neutralizing SARS-CoV-2 antibodies. However, 51 out of the 2994 blood samples from individual subjects were positive with regard to the neutralizing antibodies and at the same time negative for S1 IgG antibodies, and 112 out of the 2994 blood samples from individual subjects were negative with regard to the neutralizing antibodies and at the same time positive for S1 IgG antibodies (Figure 1). Those patients being negative with regard to the neutralizing antibodies and at the same time positive for S1 IgG antibodies are more likely to be female, whereas patients with positive detection of neutralizing antibodies but lacking detection of S1 IgG antibodies were younger than patients in whom both neutralizing and S1 antibodies were detectable (Table 1). When analyzing the entire study population (2994 outpatients), we saw that S1 IgG antibodies (women: 223.98 ± 3.81; men: 207.80 ± 4.59; *p* = 0.014) and neutralizing antibodies (women: 66.65 ± 0.82; men: 62.88 ± 1.01; *p* = 0.021) are higher in women than in men.

## 4. Discussion

Patients who do not develop adequate titers of S1 IgG antibodies after vaccination, either because they are very old or because they suffer from diseases such as end-stage renal failure or patients after chemotherapy for tumor diseases, have a high risk of developing severe COVID-19 disease with a potentially fatal outcome despite vaccination [12,13,14]. Our study also shows that there is a relevant number of patients who, despite developing significant titers of S1 antibodies, do not have relevant amounts of neutralizing antibody titers. Neutralizing antibodies usually bind to proteins on the surface of a cell of the pathogen (in the case of bacteria and fungi) or on the viral surface in the case of viruses and either sterically prevent the pathogen from binding to the host cell or prevent the conformational change of the proteins, which is necessary for entry into the host cell. Thus, the antibodies can prevent infection and possible damage by the pathogen without the need to recruit cells of the immune system. Only some of the antibodies formed after infection or vaccination and binding to the pathogen have a neutralizing effect. Non-neutralizing antibodies bind to the pathogen but do not have a neutralizing effect, but use other functions of antibodies, such as opsonization and activation of the complement system, to remove the pathogen [15,16,17].

We suspect that these patients having too few neutralizing antibodies but enough S1 antibodies do not have adequate protection against symptomatic future SARS-CoV-2 infections, although this remains to be demonstrated in clinical trials. It is also interesting to note that a few patients have high titers of neutralizing antibodies without having relevant titers of S1 antibodies. The main neutralizing antibodies are IgG antibodies against the S1 protein of the virus because this protein is the molecular receptor to the host’s ACE2 protein and is hence key for virus entry into human cells. However, other neutralizing antibodies do also exist—for example, other immunoglobulin classes or antibodies against other virus proteins such as the nucleocapsid protein. However, to the best of our knowledge, there is no study with an adequate sample size published so far indicating that neutralization of the SARS-CoV-2 virus occurs without the presence of neutralizing S1 IgG antibodies. The underlying molecular reasons need to be explored. Using modern molecular tools such as single-cell RNA sequencing of tissues where the immune response is triggered after infection such as biopsy materials from the upper respiratory tract system might be useful here. This strategy led to the discovery of a distinct inflammatory predisposition of different immune cell subtypes relevant to COVID-19 in patients with hypertension that correlated with critical disease progression but was already present before SARS-CoV-2 infection, i.e., in SARS-CoV-2-negative patients with hypertension. Moreover, immune activation in hypertensive patients was largely augmented under COVID-19, providing a novel potential explanation for the adverse course of the disease related to a hyperinflammatory response in these patients with cardiovascular disease [18,19], Single-cell RNA sequencing hence might be a favorable tool to characterize the underlying molecular pathways in patients with developing adequate levels of IgG antibodies against the S1 protein but inadequate levels of neutralizing SARS-CoV-2 virus antibodies. Our limited clinical data indicated that those patients being negative with regard to the neutralizing antibodies and at the same time clearly positive for S1 IgG antibodies are more likely to be female, whereas age—a well-known clinical risk factor deterring humoral and cellular immune response after SARS-CoV-2 infection or vaccination [20,21]—was quite similar in both groups. Given our finding that those patients with the unusual response with regard to neutralizing antibodies, but clearly positive for S1 IgG antibodies are more likely to be female, it is of note that estrogen regulates the expression of SARS-CoV-2 receptor ACE2 in differentiated airway epithelial cells [22]. In addition, it is known that female reproductive steroids, estrogen and progesterone, and their metabolite allopregnanolone, are anti-inflammatory, reshape the competence of immune cells, stimulate antibody production, and promote proliferation and repair of respiratory epithelial cells [23,24]. It remains to be analyzed in more detail whether or not these sex-specific differences in the immune response to SARS-CoV-2 infection or vaccination may be also important to understand better our key findings. Sex-dependent effects of either infection or vaccination with regard to the humoral and cellular responses have been reported [25]. Sex can affect the innate and adaptive immune system responses, predisposition to autoimmunity, and vaccine efficacy [26,27]. This difference could be linked to sex steroid hormone concentrations [28]. Our finding would suggest that not just the quantity of the humoral immune response to SARS-CoV-2 infection but also the quality of the immune response might be sex dependent. The underlying molecular mechanisms, however, are yet unknown.

In any case, based on our findings, we suggest that monitoring the humoral immune response after SARS-CoV-2 infection or vaccination should include measurements of neutralizing antibodies as well as S1 antibodies to ensure proper analysis of the humoral immune response. Subjects with adequate levels of IgG antibodies against the S1 protein but no neutralizing SARS-CoV-2 virus antibodies might be a high-risk population for (re-)infection. This hypothesis, however, needs to be proven in adequately powered clinical studies.

Our study also has limitations; first and most important, we do not have clinical follow-up data after blood collection, and hence we cannot finally prove that patients developing adequate levels of IgG antibodies against the S1 protein but inadequate levels of neutralizing SARS-CoV-2 virus antibodies are at higher risk for infection and adverse clinical outcome. This study was conducted at the Institute of Clinical Chemistry (IMD) Berlin). This has the advantage that we were able to investigate a very large number of cases—several times larger than in comparable studies. At the same time, however, we only have a limited number of clinical data on patients. We only received information about age and gender from the referring physicians. Finally, we used a method to evaluate neutralizing antibodies (ACE2-competition binding assay) that mainly detects RBD-specific antibodies. There are actually neutralizing antibodies reported that bind to NTD or S2 that do not block the ACE2 binding region of the S1 subunit of the S protein of the virus.

## Figures and Tables

**Figure 1 microorganisms-10-02067-f001:**
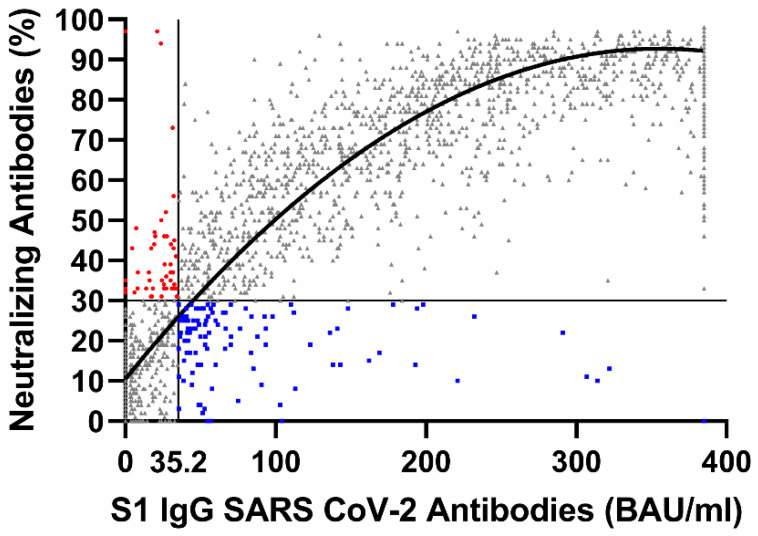
Non-linear relationship (Y = 9.994 + 0.4481X − 0.0006096X^2^, R^2^ = 0.8772) between S1 IgG antibodies and neutralizing SARS-CoV-2 virus antibodies. The lines in the figure show the threshold values of both methods according to the methods manuals [5,6] at which the tests are to be considered clearly positive. In total, 51 out of the 2994 blood samples from individual subjects were positive with regard to the neutralizing antibodies and at the same time negative for S1 IgG antibodies (red dots), whereas 112 out of the 2994 blood samples from individual subjects were negative with regard to the neutralizing antibodies and at the same time positive for S1 IgG antibodies (blue dots). S1 IgG: S1 IgG SARS-CoV-2 Antibodies (BAU/mL); Neutralizing Antibodies: Neutralizing SARS-CoV-2 Antibodies (%).

**Table 1 microorganisms-10-02067-t001:** Characteristics of the study cohort.

Parameters	Group 1 (*n* = 51)	Group 2 (*n* = 112)	Group 3 (*n* = 2311)	Group 4 (*n* = 520)
Age, mean (SD), y	50.0(15.43) *	58.0(18.52)	56.2(15.99)	50.6(17.85) ***
Male, No. (%)	20(47.6)	39(46.4) *	1233(59.6)	253(54.3)
S1 IgG antibodies, mean (SD), BAU/ml	21.9(10.18) ***	83.4(67.91) ***	274.8(121.18)	7.1(9.76) ***
Neutralizing antibodies, mean (SD), %	42.0(15.75) ***	19.6(8.26) ***	79.8(18.92)	9.8(8.40) ***

S1 IgG: S1 IgG SARS-CoV-2 Antibodies (BAU/mL); Neutralizing Antibodies: Neutralizing SARS-CoV-2 Antibodies (%). Group 1: S1 IgG Antibodies < 35.2 (BAU/mL) and Neutralizing Antibodies > 30 (%); Group 2: S1 IgG Antibodies > 35.2 (BAU/mL) and Neutralizing Antibodies < 30 (%); Group 3: S1 IgG Antibodies > 35.2 (BAU/mL) and Neutralizing Antibodies > 30 (%); Group 4: S1 IgG Antibodies < 35.2 (BAU/mL) and Neutralizing Antibodies < 30 (%). Continuous parameters were compared by one-way ANOVA; Sex comparison was undertaken by Chi-Square test. *: *p* < 0.05; ***: *p* < 0.0001 compared to group 3.

## Data Availability

The data sets of this study are available on reasonable request from the corresponding authors.

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
