# Peer review of "Outliers Matter—Correlation between S1 IgG SARS-CoV-2 Antibodies and Neutralizing SARS-CoV-2 Antibodies"

_microorganisms, 2022, doi:10.3390/microorganisms10102067_

Round 1
Reviewer 1 Report
This is a very complete and interesting manuscript.
This manuscript examines the risk of (re)infection with COVID-19 and the two types of antibodies produced by COVID-19 infection or COVID-19 vaccination. Simultaneous quantitative measurements of S1 and neutralizing antibodies are performed to verify their association with the risk of (re)infection. "There is a relevant number of patients who, despite developing significant titers of S1 antibodies, do not have relevant amounts of neutralizing antibody titers probably being at high risk of (re)infection," is a new and very messaged conclusion. And we believe that the process of leading to that conclusion was scientifically valid. This conclusion is a take home message that both S1 and neutralizing antibodies need to be measured together to determine vaccine efficacy. And this conclusion suggests that the commonly used S1 IgG antibody assay alone is not sufficient. It will be an important manuscript for future discussions on how to determine the efficacy of the COVID-19 vaccination.
One minor point is that the statistical values in the Table should be written to the correct number of significant digits.
Age should be written to one decimal place as in the text.
Percentages of neutralizing antibodies should also be to one decimal place.
Author Response
Reviewer 1
This is a very complete and interesting manuscript.
This manuscript examines the risk of (re)infection with COVID-19 and the two types of antibodies produced by COVID-19 infection or COVID-19 vaccination. Simultaneous quantitative measurements of S1 and neutralizing antibodies are performed to verify their association with the risk of (re)infection. "There is a relevant number of patients who, despite developing significant titers of S1 antibodies, do not have relevant amounts of neutralizing antibody titers probably being at high risk of (re)infection," is a new and very messaged conclusion. And we believe that the process of leading to that conclusion was scientifically valid. This conclusion is a take home message that both S1 and neutralizing antibodies need to be measured together to determine vaccine efficacy. And this conclusion suggests that the commonly used S1 IgG antibody assay alone is not sufficient. It will be an important manuscript for future discussions on how to determine the efficacy of the COVID-19 vaccination.
One minor point is that the statistical values in the Table should be written to the correct number of significant digits.
- Age should be written to one decimal place as in the text.
Response: We did as suggested.
- Percentages of neutralizing antibodies should also be to one decimal place.
Response: We did as suggested.
Reviewer 2
The authors analyzed the serum IgG profile from 2994 subjects and found there was a positive correlation between SARS-CoV-2 S1-specific IgG and neutralizing antibodies. However, the methods used in this paper have limitations that compromised the interpretation of the results. In addition, lots of previous studies have reported similar findings and this paper failed to provide novel insights based on the data. Overall, I don’t think this paper meets the standard of the journal.
- There are lots of typos and grammar/format mistakes in this manuscript, please carefully proofread the paper.
Response: We corrected spelling errors.
The authors used ACE2 competition binding ELISA to evaluate the neutralization potency of the antibodies. However, not all neutralizing antibodies bind to spike RBD. There are neutralizing antibodies that don’t compete with ACE2, which could be the cases in group 1.
Response: We fully agree that there are neutralizing antibodies that don’t compete with ACE2. We thus added a general para to the discussion describing what neutralizing antibodies are: Neutralizing antibodies usually bind to proteins in the surface of a cell of the pathogen (in the case of bacteria and fungi) or on the viral surface in the case of viruses and either sterically prevent the pathogen from binding to the host cell or prevent the conformational change of the proteins, which is necessary for entry into the host cell. Thus, the antibodies can prevent infection and possible damage by the pathogen without the need to recruit cells of the immune system. Only some of the antibodies formed after infection or vaccination and binding to the pathogen have a neutralizing effect. Non-neutralizing antibodies bind to the pathogen but do not have a neutralizing effect, but use other functions of antibodies, such as opsonization and activation of the complement system, to remove the pathogen. (line 113-122)
The main neutralizing antibodies are IgG antibodies against the S1 protein of the virus, because this protein is the molecular receptor to the host`s ACE2 protein and is hence key for virus entry of human cells. However, other neutralizing antibodies do also exist – for example other immunoglobulin classes or antibodies against other virus proteins such the nucleocapsid protein. However, to the best of our knowledge, there is no study with adequate sample size published so far indicating that neutralization of the SARS COV-2 virus occurs without presence of neutralizing S1 IgG antibodies. We added this para to the discussion as well to address your point.
- S1-specific antibodies can be non-neutralizing due to immune escape of SARS-CoV-2 variants, which could be the cases in group 2. However, it was not clear which spike variant was used for ELISA in this paper.
Response: Yes, we fully agree that it is important to note which variant were detected by our neutralizing test. Our test (according to the manufacturer information’s and internal validations) recognizes neutralization that against the wild type variant as well as against the alpha, beta, and delta variant of the virus, but not against the omicron variant of the virus. We added this information to the method section. The omicron variant did not exist when the study was done (study period: April 1st 2021 to August 30th 2021). Therefore, all virus variants playing a substantial role in the pandemic of COVID-19 during the study period in Germany were covered.
Reviewer 3
In the submitted brief report “Outliers matter - Correlation between S1 IgG SARS CoV-2 Antibodies and neutralizing SARS CoV-2 Antibodies” The authors has reported some interesting findings about the correlation and outliers between S1IgG and SARS-coV-2 neutralizing antibodies by taking around 3000 samples.
The report is quite interesting and may be interesting to researchers in the field of virology and infectious diseases. The findings of the study are intriguing and has open the research to answer various questions about SARS-CoV-2. The manuscript is well written and can be accepted for publication with minor revisions to address the following concerns:
- Please provide more details of the samples used for the study. For examples, their vaccination status, whether the patients had prior SARS-CoV-2 infections, age groups, genders.
Response: This study was conducted at an Institute of Clinical Chemistry. This has the advantage that we were able to investigate a very large number of cases - several times larger than in comparable studies. At the same time, however, we only have a limited number of clinical data on patients. We only received information about age and gender from the referring physicians. Unfortunately, we did not have any further clinical data. We do not know for example whether the patients were vaccinated or had been infected. We mentioned this as a study limitation.
- Minor spelling check on line 61.
Response: We corrected spelling errors.
- The authors provide the finding that one set of outliers (S1 IgG positive and Neutralizing Ab -ve) are more likely to be female. This is quite interesting. Can author provide some information on these samples such as their ages. Was it observed in older females or younger groups? This information might help in some explanation about estrogen role.
Response: These information’s were given in the table. See information’s for group 2. in the table. In addition, we analysed the whole study population for potential sex differences and added these data to the results section as follows: When we were analysing the entire study population (2994 outpatients), we saw that S1 IgG antibodies (women: 223.98±3.81; men: 207.80±4.59; p=0.014) and neutralizing antibodies (women: 66.65±0.82; men: 62.88±1.01; p=0.021) are higher in women than in men.
- Can the author provide some speculative reasons why this outlier is more pronounce in females?
Response: So far there is no clear answer to your important question, to address your point we added the following para to the discussion: Sex-dependent effects to either infection or vaccination with regard to the humoral and cellular have been reported. Sex can affect the innate and adaptive immune system responses, predisposition to autoimmunity, and vaccine efficacy. This difference could be linked to sex steroid hormones concentrations. Our finding would suggest that not just the quantity of the humoral immune response to SARS CoV-2 infection but also the quality of the immune response might be sex dependent. The underlying molecular mechanism, however, are yet unknown. (line 161-168)
- In the discussion part, provide some explanation of what neutralizing ab are and how it is different than the S1IgG. Regarding the SARS-CoV-2 infection, which antibody is more important in protection?
Response: Yes, this an important question, we hence added the following para: Neutralizing antibodies usually bind to proteins in the surface of a cell of the pathogen (in the case of bacteria and fungi) or on the viral surface in the case of viruses and either sterically prevent the pathogen from binding to the host cell or prevent the conformational change of the proteins, which is necessary for entry into the host cell. Thus, the antibodies can prevent infection and possible damage by the pathogen without the need to recruit cells of the immune system. Only some of the antibodies formed after infection or vaccination and binding to the pathogen have a neutralizing effect. Non-neutralizing antibodies bind to the pathogen but do not have a neutralizing effect, but use other functions of antibodies, such as opsonization and activation of the complement system, to remove the pathogen. (line113-122)
- Please provide some information in the scenario, People with high level of Neutralizing Ab but negative for S1 IgG. Do they have protection against SARS-CoV-2 reinfection?
Response: This is a very important question. Since we have no follow-up data, this is basically unknown. We mentioned this important topic in the manuscript as follows in the study limitation section: “first and most important, we do not have clinical follow-up data after blood collection, hence we cannot finally prove that patients developing adequate levels of IgG antibodies against the S1 protein but inadequate levels of neutralizing SARS CoV-2 virus antibodies are at higher risk for infection and adverse clinical outcome.”
Reviewer 2 Report
The authors analyzed the serum IgG profile from 2994 subjects and found there was a positive correlation between SARS-CoV-2 S1-specific IgG and neutralizing antibodies. However, the methods used in this paper have limitations that compromised the interpretation of the results. In addition, lots of previous studies have reported similar findings and this paper failed to provide novel insights based on the data. Overall, I don’t think this paper meets the standard of the journal.
· There are lots of typos and grammar/format mistakes in this manuscript, please carefully proofread the paper.
· The authors used ACE2 competition binding ELISA to evaluate the neutralization potency of the antibodies. However, not all neutralizing antibodies bind to spike RBD. There are neutralizing antibodies that don’t compete with ACE2, which could be the cases in group 1.
· S1-specific antibodies can be non-neutralizing due to immune escape of SARS-CoV-2 variants, which could be the cases in group 2. However, it was not clear which spike variant was used for ELISA in this paper.
Author Response

(The authors gave the same response as above.)

Reviewer 3 Report
In the submitted brief report “Outliers matter - Correlation between S1 IgG SARS CoV-2 Antibodies and neutralizing SARS CoV-2 Antibodies” The authors has reported some interesting findings about the correlation and outliers between S1IgG and SARS-coV-2 neutralizing antibodies by taking around 3000 samples.
The report is quite interesting and may be interesting to researchers in the field of virology and infectious diseases. The findings of the study are intriguing and has open the research to answer various questions about SARS-CoV-2. The manuscript is well written and can be accepted for publication with minor revisions to address the following concerns:
1. Please provide more details of the samples used for the study. For examples, their vaccination status, whether the patients had prior SARS-CoV-2 infections, age groups, genders.
2. Minor spelling check on line 61.
3. The authors provides the finding that one set of outliers (S1 IgG positive and Neutralizing Ab -ve) are more likely to be female. This is quite interesting. Can author provide some information on these samples such as their ages. Was it observed in older females or younger groups? This information might help in some explanation about estrogen role.
4. Can the author provide some speculative reasons why this outlier is more pronounce in females?
5. In the discussion part, provide some explanation of what neutralizing ab are and how it is different than the S1IgG. Regarding the SARS-CoV-2 infection, which antibody is more important in protection?
6. Please provide some information in the scenario, People with high level of Neutralizing Ab but negative for S1 IgG. Do they have protection against SARS-CoV-2 reinfection?

Author Response

(The authors gave the same response as above.)

Round 2
Reviewer 2 Report
1. There are still lots of typos in the paper, some of them are listed here. Please also correct errors with punctuation marks and spaces.
Line 47 "sample" - "samples", "methos" - "methods"
Line 113 "in the surface" - "on the surface"
Line 124 "neuralising" - "neutralizing"
Line 131 "exis" - "exist"
Line 135 "needs" - "need"
Line 165 "hormones" - "hormone", "quatity" - "quantity"
Line 166 "imme" - "immune"
Line 167 "Theunderlying" - "The underlying"
2. Maybe I didn't make my point clear in the previous comment. The spike protein consists of 2 subunits - S1 and S2, and the S1 subunit contains the receptor binding domain (RBD) and N-terminal domain (NTD). The method used to evaluate neutralizing antibodies in this paper (ACE2-competition binding assay) mainly detects RBD-specific antibodies. There are actually many neutralizing antibodies reported that bind to NTD or S2 that don't block ACE2 binding. Please make it clear in the paper that the neutralizing antibodies you detected are RBD-specific, but not S1-specific. And this would be another limitation that should be included in the discussion.
Author Response
Please see the attachment. Since I could not separately upload the revised version of our paper and the response letter to reviewer 2 (your system does not allow this, you should improve it) , I combined the response letter and the manuscript to on file.
